# Genomic evidence of contemporary hybridization between *Schistosoma* species

**Duncan J. Berger** [1,2�]*, **Elsa Léger**[2,3�]*, **Geetha Sankaranarayanan**[1], **Mariama Sène**[4], **Nicolas D. Diouf**[4], **Muriel Rabone**[5], **Aidan Emery**[5], **Fiona Allan**[5,6], **James A. Cotton**[1,3¤a], **Matthew Berriman**[1,3¤b‡]*, **Joanne P. Webster**[2,3‡]*

**1** Wellcome Sanger Institute, Hinxton, United Kingdom, **2** Royal Veterinary College, University of London, London, United Kingdom, **3** London Centre for Neglected Tropical Diseases Research, Imperial College Faculty of Medicine, London, United Kingdom, **4** Unité de Formation et de Recherche des Sciences Agronomiques, d'Aquaculture et de Technologies Alimentaires, Université Gaston Berger, Saint-Louis, Senegal, **5** The Natural History Museum, Department of Life Sciences, Cromwell Road, London, United Kingdom, **6** Pelagic Ecology Research Group, Scottish Oceans Institute, Gatty Marine Laboratory, School of Biology, University of St Andrews, St Andrews, United Kingdom

These authors contributed equally to this work.
¤a Current address: Institute of Biodiversity, Animal Health and Comparative Medicine, Wellcome Centre for Integrative Parasitology, University of Glasgow, Glasgow, United Kingdom
¤b Current address: Institute of Infection, Immunity and Inflammation, University of Glasgow, Glasgow, United Kingdom
‡ Authors jointly supervised this work.
* duncan.berger@bath.edu (DB); leger.elsa@gmail.com (EL); mb4@sanger.ac.uk (MB); jowebster@rvc.ac.uk (JPW)

**Data Availability Statement:** Raw Illumina sequencing data generated during this study are available in the European Nucleotide Archive (ENA) repository under study accession number

## Abstract

Hybridization between different species of parasites is increasingly being recognised as a major public and veterinary health concern at the interface of infectious diseases biology, evolution, epidemiology and ultimately control. Recent research has revealed that viable hybrids and introgressed lineages between *Schistosoma* spp. are prevalent across Africa and beyond, including those with zoonotic potential. However, it remains unclear whether these hybrid lineages represent recent hybridization events, suggesting hybridization is ongoing, and/or whether they represent introgressed lineages derived from ancient hybridization events. In human schistosomiasis, investigation is hampered by the inaccessibility of adult-stage worms due to their intravascular location, an issue which can be circumvented by post-mortem of livestock at abattoirs for *Schistosoma* spp. of known zoonotic potential. To characterise the composition of naturally-occurring schistosome hybrids, we performed whole-genome sequencing of 21 natural livestock infective schistosome isolates. To facilitate this, we also assembled a *de novo* chromosomal-scale draft assembly of *Schistosoma curassoni*. Genomic analyses identified isolates of *S. bovis*, *S. curassoni* and hybrids between the two species, all of which were early generation hybrids with multiple generations found within the same host. These results show that hybridization is an ongoing process within natural populations with the potential to further challenge elimination efforts against schistosomiasis.

PRJEB22769. The *S. curassoni* reference assembly and associated raw Hi-C reads are in the ENA repository under study accession number PRJEB44848 and were deposited in Zenodo (https://zenodo.org/record/5154271). The raw PacBio subreads are in the ENA repository under study accession number PRJEB3054. The code to reproduce all analysis and figures for this manuscript is described in https://github.com/duncanberger/schisto_livestock_hybrids.

**Funding:** This work was supported by the Wellcome Trust (grant number 206194); the Biotechnology and Biological Sciences Research Council, the Department for International Development, the Economic & Social Research Council, the Medical Research Council, the Natural Environment Research Council and the Defence Science & Technology Laboratory, under the Zoonoses and Emerging Livestock Systems (ZELS) programme (grant number BB/L018985/1 to JPW and MS and grant number BB/S013822/1 to JPW, MS and NDD). FA, AE and MR received funding from the Wellcome Trust (grant number 104958/Z/14/Z). The funders had no role in study design, data collection and analysis, decision to publish, or preparation of the manuscript.

**Competing interests:** The authors have declared that no competing interests exist.

## Author summary

Schistosomiasis is a chronic and debilitating major neglected tropical disease affecting both humans and livestock. Increasingly, zoonotic spillover of livestock infections, facilitated by hybridization between different *Schistosoma* species, is increasingly being recognised as a risk to human health. Multiple surveys conducted within endemic regions have found a high prevalence of these hybrid lineages. However, it is often unclear whether these lineages are derived from recent hybridization events, suggesting hybridization is ongoing and may be linked to anthropogenic environmental change, or simply indicators of introgression from ancient hybridization events. To understand the origin and evolution of these hybrid lineages, we produced a chromosomal-scale assembly of *Schistosoma curassoni* and performed whole-genome sequencing of 21 natural livestock-infective *S. curassoni*, *S. bovis* and hybridized schistosome isolates, including multi-stage sampling from the same hosts. Our analyses exclusively identified early generation hybrid lineages, including multiple unrelated generations within the same hosts, suggesting that these hybrids are viable and derived from multiple independent hybridization events.

## Introduction

Interspecific hybridization, the interbreeding of two individuals from different species, is widely recognized as an important and widespread evolutionary process [1,2]. Hybridization can facilitate the transfer of adaptive advantages across species boundaries and act as a source of genetic diversity across short time scales, facilitating rapid adaptation to new and changing environments [3–7]. While the ecological outcomes of hybridization can be highly variable, hybridization between parasites is an emerging major public and veterinary health concern [8]. Hybrid parasite lineages have been identified with, for example, increased virulence [9], reduced drug susceptibility [9] and increased host range [10,11]. It is expected that the shifting distributions of humans, livestock and parasites due to migration, trade and climate change will increase the frequency of mixed-species co-infections providing more frequent opportunities for hybridization [12–15].

Hybridization between species of schistosomes, the dioecious trematodes that cause the chronic and debilitating neglected tropical disease (NTD) schistosomiasis, is a particular public and veterinary health risk [16]. Schistosomiasis currently infects over 220 million people across 78 countries [17], and untold millions of livestock [18,19]. WHO targets aim to eliminate schistosomiasis as a public health problem in all endemic countries and interrupt transmission in selected regions by 2030 [20,21]. The recently launched WHO 2021–2030 road map states, to achieve these goals, actions required include a better understanding of zoonotic transmission, hybridization and a consideration of potential veterinary and One Health interventions [21]. Domestic livestock often play a key role in cross-species transmission of zoonoses in general, due to their close association with human populations. Furthermore, animal schistosomiasis represents a double burden to poor rural communities for whom their livestock are a crucial source of income and economic security [19].

While the epidemiological risks of *Schistosoma* spp. hybridization are only just beginning to be realised, it is becoming clear that this has the potential to alter transmission and morbidity dynamics, circumventing current control measures. For example, field and experimental studies, most notably within the Haematobium clade, have demonstrated that hybridization results in viable offspring with altered phenotypes such as that of shifted cercarial shedding patterns [22–24], increased growth rate and higher reproductive output [25,26], altered

morbidity profiles induced within their human hosts [27], as well as expanded intermediate or definitive host ranges [23,28–30] relative to their single species counterparts. Indeed, recently a child in Niger was found to be infected with a schistosome displaying markers from two live-stock-infective schistosome species, *S. bovis* and *S. curassoni*, which on their own do not reach patency, further indicating that hybridization could promote expanded host range with zoonotic transmission [30]. Likewise, countries, most notably those across West Africa, which can have high proportions of hybridized schistosomes amongst both their human and livestock populations, often display extremely high persistence of infection prevalence and intensities despite years of targeted control within the human populations [20,31]. Moreover, it has been suggested that hybridization between *S. bovis* and *S. haematobium* facilitated a 2013 outbreak of schistosomiasis in Corsica (France), outside the endemic zone of these species [32–34], although the classification of these samples as recent hybrids has been disputed [35]. Understanding the disease dynamics and evolutionary consequences of these hybridization events in both humans and livestock is therefore essential for predicting disease persistence and to inform control efforts, as well as broaden our knowledge regarding inter-specific hybridization events amongst parasites and pathogens in general.

Schistosomes derived from possible hybridization events were originally identified using egg morphology [36–38] or biochemical markers [39,40]. More recently, single-gene or multi-gene sequencing has been used to confirm their existence, in particular through identifying cases of mitochondrial and nuclear discordance or nuclear markers heterozygous for alleles typical of different species [31,41–43]. While these approaches have identified that hybridization could be common in natural populations, they lack the precision to reveal the full history of hybrid populations and may miss highly backcrossed or introgressed hybrid lineages. A recent genomic analysis of hybridization in schistosomes reported no evidence for contemporary hybridization amongst samples from infected children from Niger (*n* = 96), instead finding evidence of an ancient introgression event from the livestock-infective *S. bovis* into *S. haematobium* that occurred an estimated 108–613 generations ago [44]. Both this study and a second smaller-scale genomic survey (*n* = 19) found evidence of directional selection on the *S. bovis*-derived alleles in a gene that might facilitate definitive host infection [35,44]. Recent larger-scale genotyping surveys have also suggested that both ancient (occurring hundreds of generations prior to sampling) and more recent hybrid lineages could be present in schistosome populations [31,42,45].

These studies leave, however, many unresolved questions about the details of hybridization in current schistosome populations. These details matter: for example, the presence of first-generation hybrids does not demonstrate that hybrids are themselves fertile and of long-term importance, while fertile back-crosses with parental species could lead to introgression of clinically important traits between species. Evidence of adaptive introgression from historic hybridization can identify adaptive alleles, whereas evidence of recent hybridization could indicate demographic shifts or anthropogenic environmental changes influencing parasite population structures.

Here we present novel whole-genome sequence data from 21 schistosome isolates from naturally-infected cattle, collected from two regions of Northern Senegal where livestock are known to be infected with *S. curassoni*, *S. bovis* and hybrids between these species [31,46]. Sequenced isolates were selected to represent both non-hybrid genotypes of both species and hybrid genotypes found in this population, and to include both miracidial and adult stages. By sampling from livestock, we were uniquely able to capture multiple developmental stages, notably both adult pairs and miracidia representing parents and potential offspring, respectively. These stages are otherwise unavailable to all previous studies on humans from which adult schistosomes cannot be obtained, even following chemoexpulsion, in contrast to the

situation for many other helminth species. We used these data to screen for evidence of hybridization and determine whether admixed samples were derived from recent hybridization events or whether these represent introgressed lineages derived from ancient hybridization events.

## Results

### A *de novo* chromosomal-scale *Schistosoma curassoni* reference genome

To facilitate our population analyses we generated a *de novo* assembly for the livestock parasite, *S. curassoni*. Using PacBio continuous long reads and Arima High-throughput Chromosome Conformation Capture (Hi-C) reads, we produced a chromosomal-scale primary assembly with a total length of 397.0 Mb and 94.8% of the assembly comprised 8 chromosomal scaffolds (scaffold N50 = 46.0 Mb, Fig 1A and 1B). The assembly has similar contiguity and completeness to the *S. mansoni* (version 9) reference genome [47] and is vastly more contiguous than

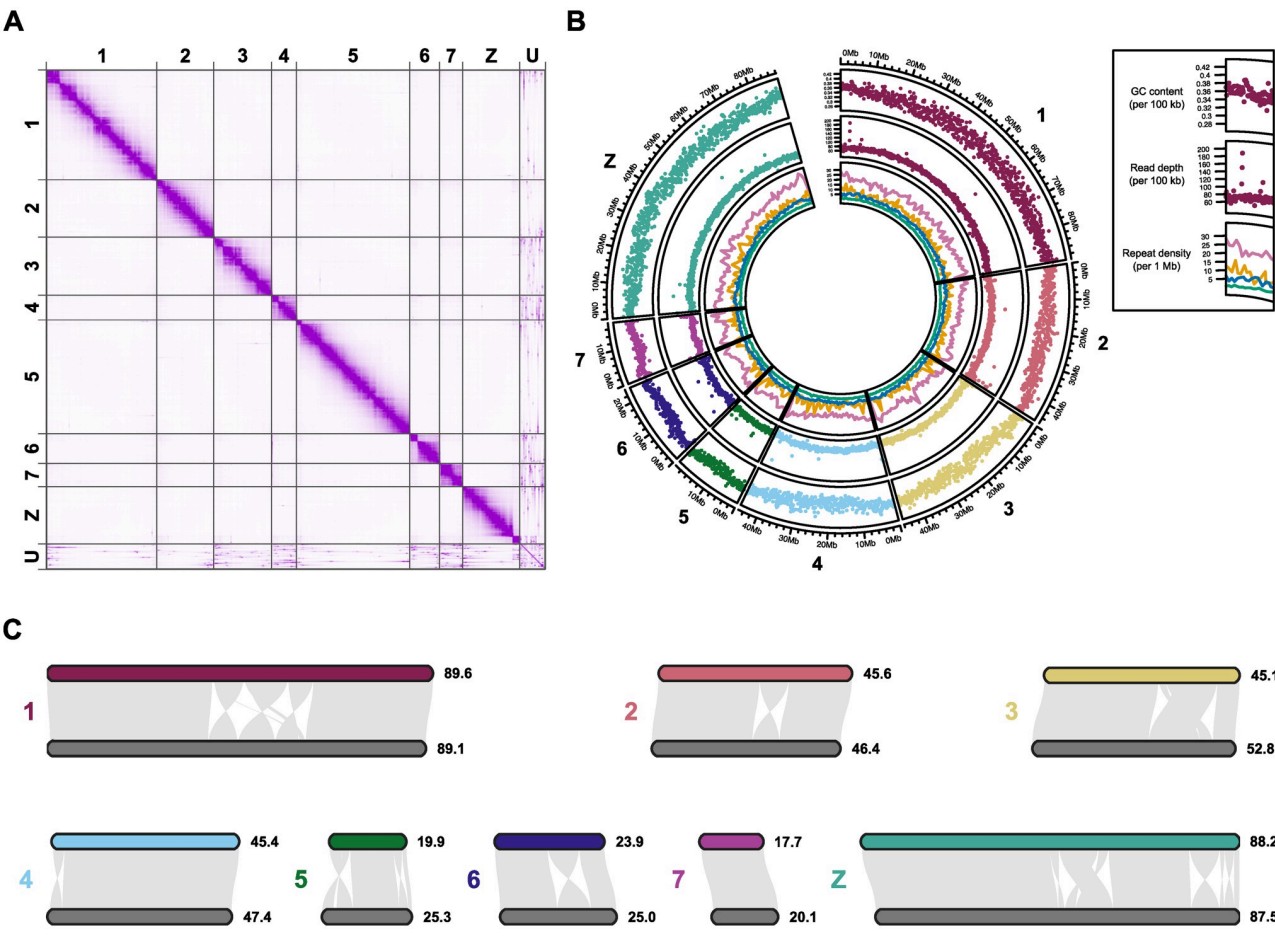

**Fig 1. Chromosomal synteny between *Schistosoma mansoni* (v9) and *Schistosoma curassoni* (V1).** (A) Hi-C contact map of the chromosomes (1–7, Z) and unplaced contigs/scaffolds (U, *n* = 361), visualised using Juicebox Assembly Tools. (B) Circos plot of genome-wide features. Outer: median GC content, shown as a proportion of 100 kb windows along each chromosome, middle: depth of coverage of PacBio subreads (median of 100 kb windows), inner: density of annotated repeat elements in 1 Mb windows, including LINEs (pink), DNA (green), LTRs (orange) and Penelope (blue). (C) We used MCscan to identify homologous chromosomal regions between our newly sequenced *S. curassoni* assembly and the *S. mansoni* (v9) assembly. Per-chromosome synteny showing syntenic paths, structural rearrangements (inversions, translocations and duplications) and un-aligned regions between *S. curassoni* chromosomes (shown in different colours) and *S. mansoni* (chromosomes shown in dark grey). Chromosomal lengths (in Mb) are shown at the end of each chromosome.

**Table 1. Genome assembly summary statistics.** We assessed the completeness of the curated assembly by searching for conserved, single copy genes using BUSCO (Benchmarking Universal Single-Copy Orthologs) and the eukaryota lineage (odb10) dataset.

| | | *S. mansoni* | *S. haematobium* | *S. japonicum* | *S. bovis* | *S. curassoni* | *S. curassoni* |
|---|---|---|---|---|---|---|---|
| | | (Smansoni_v9) | (Shae.V3) | (SjV3) | (ASM395894v1) | (S_curassoni_Dakar_0011_upd) | (Scur_626247_v1.0) |
| **Assembly statistics:** | **Assembly size (Mb)** | 391.4 | 409.9 | 408.5 | 373.5 | 344.2 | 397.0 |
| | **Assigned to chromosomes (%)** | 100 | 94.9 | 96.0 | 0 | 0 | 94.8 |
| | **Gaps (count)** | 356 | 284 | 271 | 141786 | 58187 | 365 |
| | **Scaffold number** | 11 | 563 | 337 | 4774 | 60140 | 370 |
| | **Scaffold N50 (Mb)** | 45.7 | 48.2 | 48.8 | 0.2 | <0.1 | 46.1 |
| | **Largest scaffold (Mb)** | 88.0 | 92.5 | 90.6 | 1.1 | 0.2 | 89.6 |
| **BUSCO Eukaryota:** | **Complete & single copy (%)** | 89.0 | 86.7 | 85.9 | 82.4 | 56.5 | 85.1 |
| | **Duplicated (%)** | 0.8 | 1.2 | 2.0 | 2.0 | 0.0 | 2.4 |
| | **Fragmented (%)** | 3.9 | 5.5 | 7.1 | 7.8 | 27.1 | 5.9 |
| | **Missing (%)** | 6.3 | 6.6 | 5.0 | 7.8 | 16.4 | 6.5 |
| | **Publication** | [47] | [95] | [96] | [97] | [48] | **Herein** |

previous Illumina-only genome assemblies for *Schistosoma* spp. [48]. BUSCO (Benchmarking Universal Single-Copy Orthologs) scores indicated a high completeness for a parasitic trematode species, identifying 85.1% of the complete set of single-copy orthologs in the Eukaryota lineage dataset, equivalent to other chromosomal-scale assemblies for this genus (Table 1). Whole genome alignment of our *S. curassoni* and the *S. mansoni* assemblies identified high levels of sequence conservation and limited structural variation consistent with a previous analysis between *S. mansoni* and *S. haematobium* (Fig 1C) [49].

## Whole-genome sequencing of field isolated schistosomes

We performed whole-genome sequencing of 12 adult-stage worms and 13 miracidia (larval stage schistosomes) sampled from a total of four cattle hosts in Senegal (S1 Table). Based on *cox1* and ITS-rDNA genotyping we expected these samples to consist of either pure *S. bovis*, *S. curassoni* or a hybrid of the two species (Table 2). We combined these sequence datasets with a panel of 12 publicly-available samples consisting of representative sequence datasets from seven Haematobium clade species. This included two isolates from *S. bovis* and four isolates from *S. haematobium*, the latter being phylogenetically the closest species to *S. bovis* and *S. curassoni* for which multiple whole-genome sequenced samples exist.

Alignment of sequence reads to the *S. curassoni* genome showed high per-sample mapping rates for *S. bovis*, *S. curassoni* and potential hybrid samples (S2 Table). Overall, reads mapped to the majority of the genome (median 96.5% of bases covered across one or more samples), with consistent coverage across chromosomes for all samples (S3 Table) but with substantially lower mapping across the Z-specific region of the Z chromosome due to the inclusion of both female (ZW) and male (ZZ) samples (S1–S4 Figs). The difficulty in manipulating individual parasites in larval samples, and the fact that adult female schistosomes were often partially hidden within the gynaecophoric canal of adult males, both increase the risk of same-species or same-genus contamination of samples. This could potentially mirror the patterns of heterozygosity seen in early generation hybrids. To avoid misclassifying samples as early generation hybrids we therefore removed any samples showing any evidence of heavily biased allelic

**Table 2. Non-reference samples included in our analyses.** Summary of field samples used for our analyses, not shown are the 'reference samples' (previously released are published datasets) and samples that did not pass quality control. We have also reported the results of cox1/ITS-rDNA genotyping conducted prior to sequencing.

| Sample ID | Host ID | Site, Country | Lifecycle stage | *cox1* genotype | ITS-rDNA genotype | Preliminary ID |
|---|---|---|---|---|---|---|
| BK16_B3_01 | BK16_B3 | Barkedji, Senegal | Adult | *S. curassoni* | *S. curassoni* | *S. curassoni* |
| BK16_B3_02 | | | Adult | *S. bovis* | *S. bovis-S. curassoni* | Hybrid |
| BK16_B8_03 | BK16_B8 | Barkedji, Senegal | Adult | *S. curassoni* | *S. curassoni* | *S. curassoni* |
| BK16_B8_01 | | | Adult | *S. bovis* | *S. bovis-S. curassoni* | Hybrid |
| BK16_B8_02 | | | Adult | *S. bovis* | *S. bovis-S. curassoni* | Hybrid |
| BK16_B8_04 | | | Adult | *S. bovis* | *S. bovis-S. curassoni* | Hybrid |
| BK16_B8_05 | | | Adult | *S. bovis* | *S. bovis-S. curassoni* | Hybrid |
| BK16_B8_06 | | | Adult | *S. bovis* | *S. bovis-S. curassoni* | Hybrid |
| BK16_B8_08 | | | Miracidium | *S. curassoni* | *S. curassoni* | *S. curassoni* |
| BK16_B8_09 | | | Miracidium | *S. curassoni* | *S. curassoni* | *S. curassoni* |
| BK16_B8_13 | | | Miracidium | *S. curassoni* | *S. curassoni* | *S. curassoni* |
| BK16_B8_15 | | | Miracidium | *S. curassoni* | *S. curassoni* | *S. curassoni* |
| BK16_B8_17 | | | Miracidium | *S. curassoni* | *S. curassoni* | *S. curassoni* |
| BK16_B8_19 | | | Miracidium | *S. curassoni* | *S. curassoni* | *S. curassoni* |
| BK16_B8_11 | | | Miracidium | *S. bovis* | *S. bovis* | *S. bovis* |
| BK16_B8_12 | | | Miracidium | *S. bovis* | *S. curassoni* | Hybrid |
| BK16_B8_07 | | | Miracidium | *S. bovis* | *S. bovis-S. curassoni* | Hybrid |
| BK16_B8_10 | | | Miracidium | *S. bovis* | *S. bovis-S. curassoni* | Hybrid |
| BK16_B8_14 | | | Miracidium | *S. curassoni* | *S. bovis-S. curassoni* | Hybrid |
| RT15_B3_01 | RT15_B3 | Richard Toll, Senegal | Adult | NA | NA | Unknown |
| RT15_B6_01 | RT15_B6 | Richard Toll, Senegal | Adult | *S. bovis* | *S. bovis* | *S. bovis* |

depths for heterozygous variants (S5–S8 Figs and S4 and S5 Tables). In total, we removed three adult-stage samples with evidence of potential contamination (S1 Table) and also removed a single sample from Barkedji due to a low genotyping rate.

This stringent quality control thereby resulted in a high-quality variant dataset consisting of 5,421,293 single-nucleotide polymorphisms (SNPs) and 649,337 short insertions or deletions (indels) from 33 samples (19 from Barkedji Senegal, 2 from Richard Toll, Senegal, 12 previously published samples, S6 Table).

## Parasite population structure

We removed variants in strong linkage disequilibrium, resulting in a subset of 366,486 variants. Principal-component analysis (PCA) revealed separation of all reference species samples along the first four eigenvectors (Fig 2A and 2B). We found that non-reference samples were spread unevenly along the first and third eigenvectors between *S. bovis* and *S. curassoni* and phylogenetic analysis showed that samples could be divided into three distinct clades (*S. curassoni*, *S. bovis* and outgroup samples) with the remaining samples being found in an intermediate phylogenetic position between *S. curassoni* and *S. bovis* consistent with the PCA analysis (Fig 2C and S9 Fig). To determine the genetic ancestry of each sample, we used ADMIXTURE, including only the three species for which multiple samples were available: *S. curassoni*, *S. bovis* and *S. haematobium* (as well as suspected admixed samples). The lowest cross-validation error was found for K = 3 populations (S10 Fig). ADMIXTURE analysis identified all non-reference samples as belonging to either *S. curassoni*, *S. bovis* or as being admixed between the two. For the admixed samples, the proportion of *S. curassoni* ancestry varied between 0.36–

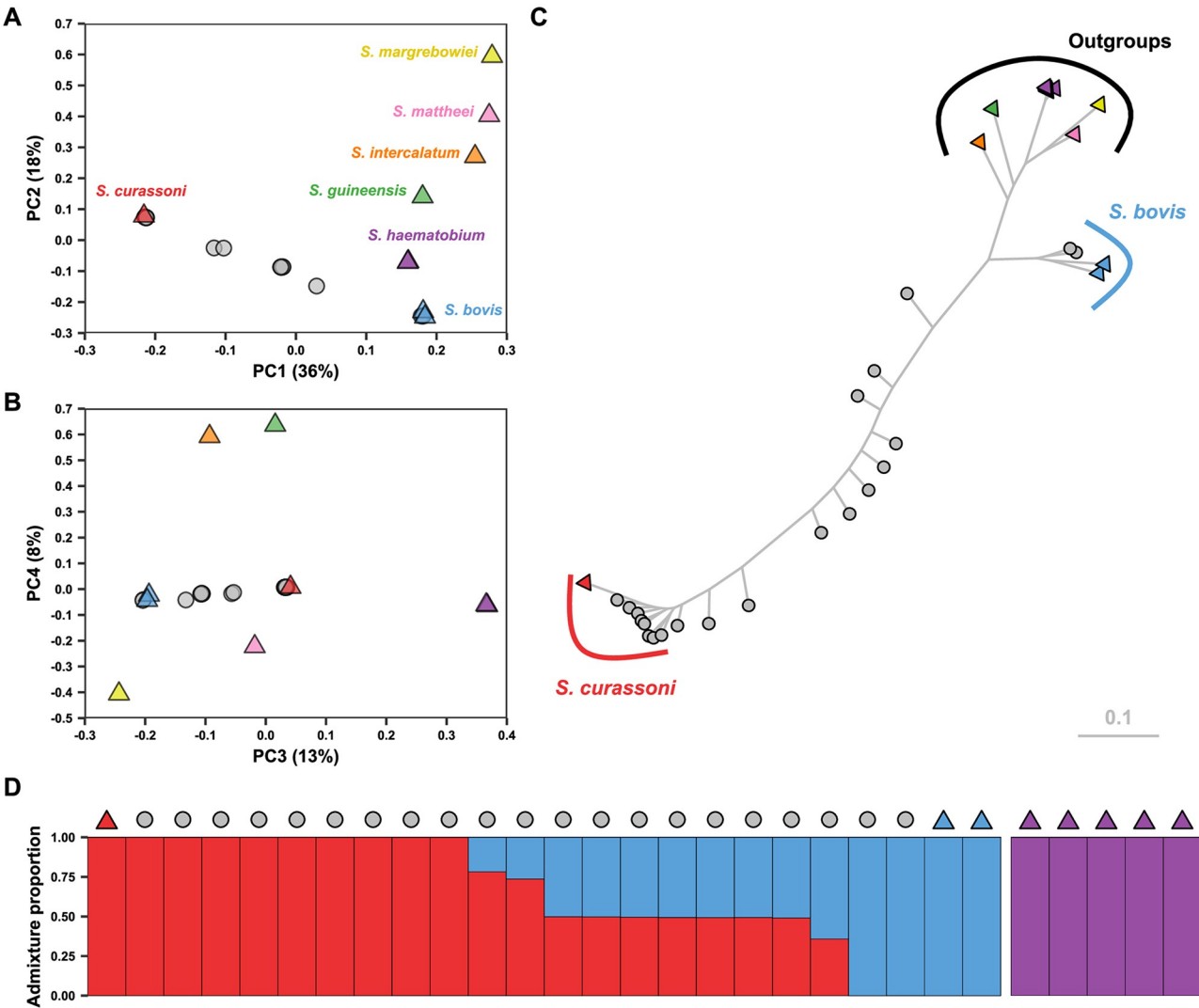

**Fig 2. Population structure of sequenced isolates.** Samples denoted with triangles represent reference samples, circles represent samples sequenced for this study. (A) Principal component analysis (PCA) of genomic differentiation between the 21 isolates sequenced for this study (grey circles) and the 12 samples included as references for each available Haematobium clade species (triangles): *S. margrebowiei* (yellow), *S. mattheei* (pink), *S. intercalatum* (orange), *S. guineensis* (green) *S. haematobium* (purple), *S. curassoni* (red) and *S. bovis* (blue). The first four principal components represented 75% of the total variance. (C) Maximum-likelihood phylogeny inferred using 23,136 parsimony informative single-nucleotide polymorphisms. Tips are coloured and shaped as in A&B, single-species and outgroups groups are highlighted. (D) ADMIXTURE plots showing the population structure, assuming three populations are present (reference samples for three species were included), we used 10-fold cross-validation and standard error estimations using 1000 bootstraps. Y-axis values show the estimated admixture proportions for each sample.

0.74 (Fig 2D and S7 Table). Analysis of the mitochondrial genome (S11 Fig) showed that all sequenced non-reference samples could be assigned as either *S. curassoni* or *S. bovis*, with 9 out of 10 admixed samples having *S. bovis* mitochondrial genotypes. Tests for kinship between all samples found no related miracidia (S8 Table).

## Genomic signatures of admixture

We designated reference populations for both *S. curassoni* (n = 3) and *S. bovis* (n = 3) selected from published datasets or those with no evidence of admixture (based on *cox1*/ITS-rDNA

and ADMIXTURE results) and generated per-sample statistics for common signatures of admixture (Fig 3). The proportion of genome-wide heterozygous variants was lowest in the *S. curassoni* samples, low in *S. bovis* samples, and highest in admixed samples (Fig 3). The most heterozygous samples were those with approximately equal contributions from the two parental species in the admixture analysis. When we selected for variants fixed between *S. curassoni* and *S. bovis* populations ($n$ = 309,040 variant sites) the proportion of heterozygous sites in admixed samples varied between 38–87% compared to only 3–4% in non-admixed samples (S9 Table).

We then used $f_3$ statistics to identify samples derived from admixture between two source populations (*S. bovis* and *S. curassoni*). $f_3$ reports the genetic divergence of a population from two reference populations and is expected to be negative only where this population has a history of admixture from these two reference populations [50]. All admixed samples had negative $f_3$ estimates suggesting gene flow between the two species (Fig 3 and Table 3). The closely-related statistic, Patterson's D (also referred to as the ABBA-BABA statistic), uses a third reference population to differentiate between gene flow (indicated by D values significantly different from zero) and incomplete lineage sorting (indicated by near zero values of D) [51]. While both $f_3$ and Patterson's D statistics are more commonly applied to populations, these have both previously been used for per sample analyses [52–55]. Using Patterson's D we identified significant gene flow between admixed samples and *S. curassoni* (range$_D$ = 0.65–0.81; range$_Z$ = 8.54–44.53) and admixed samples and *S. bovis* (range$_D$ = 0.45–0.53, range$_Z$ = 6.36–12.32; Fig 3 and Table 3).

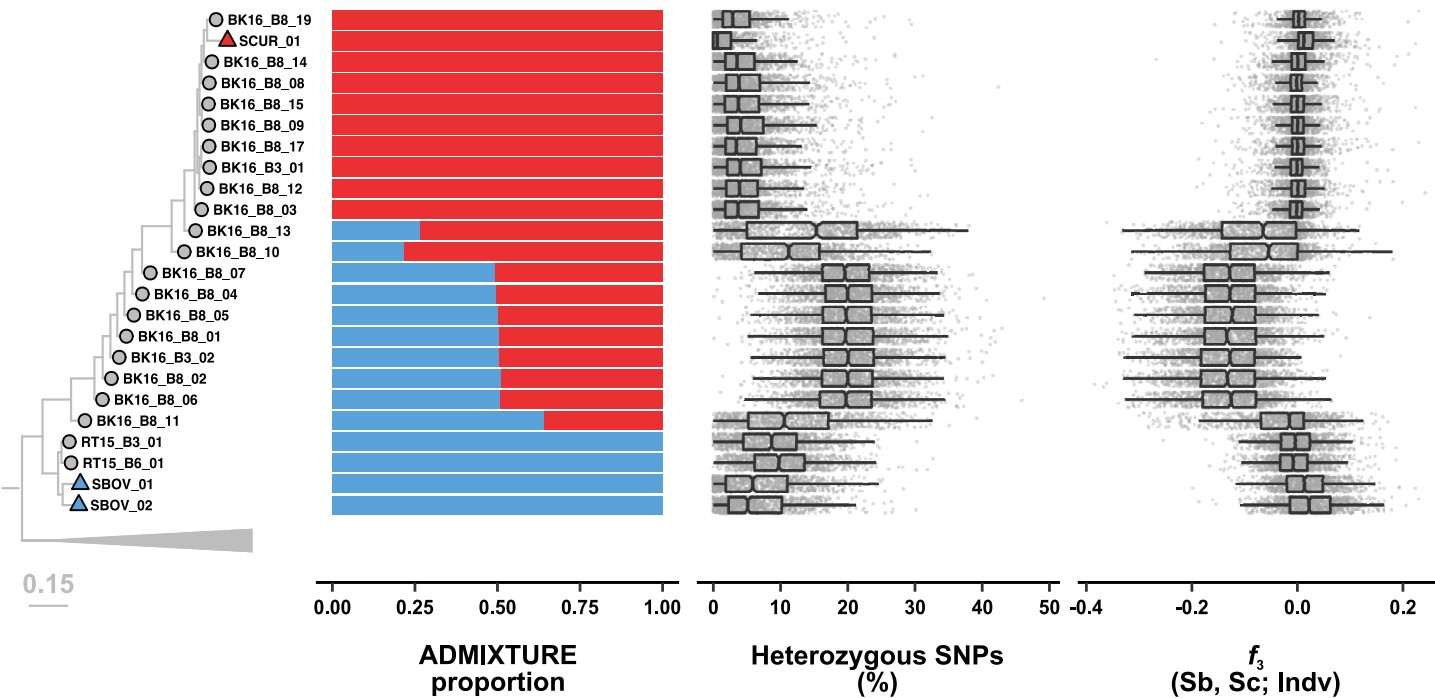

**Fig 3. Evidence of admixture in sequenced samples.** The phylogeny and admixture proportion estimates are identical to Fig 2. We calculated the proportion of heterozygous variants per 50 kb windows along each autosome. The three-population statistic ($f_3$) was used to test for admixture between each test sample and both reference populations for each species. We defined three populations described by the tree ((P1,P2),P3). We designated reference populations for *S. bovis* and *S. curassoni* as P1 and P2, respectively. Individual samples to be analysed were designated at P3.

**Table 3. Inferred species classifications and admixture statistics.**

| Sample ID | Site | Stage | Inferred species (cox1/ITS-rDNA) | Inferred species (WGS) | Inferred mitotype (WGS) | Autosomal junction count | $f_3$ (SE, Z) (Sc, Sb; Indv) | Patterson's D (Z) (Sb, Indv; Sc, Sh) | Patterson's D (Z) (Sc, Indv; Sb, Sh) |
|---|---|---|---|---|---|---|---|---|---|
| BK16_B3_01 | Barkedji | Adult | *S. curassoni* | *S. curassoni* | *S. curassoni* | 0 | -0.08 (0, -21.9) | 0.88 (61.75) | 0.00 (0.03) |
| BK16_B3_02 | Barkedji | Adult | Hybrid | $F_1$ | *S. bovis* | 1 | -0.34 (0, -299.09) | 0.72 (29.76) | 0.51 (11.33) |
| BK16_B8_01 | Barkedji | Adult | Hybrid | $F_1$ | *S. bovis* | 2 | -0.34 (0, -294.01) | 0.71 (28.77) | 0.56 (11.06) |
| BK16_B8_02 | Barkedji | Adult | Hybrid | $F_1$ | *S. bovis* | 4 | -0.34 (0, -289.7) | 0.70 (28.65) | 0.56 (10.86) |
| BK16_B8_03 | Barkedji | Adult | *S. curassoni* | *S. curassoni* | *S. curassoni* | 0 | -0.07 (0, -19) | 0.88 (62.14) | -0.01 (0.34) |
| BK16_B8_04 | Barkedji | Adult | Hybrid | $F_1$ | *S. bovis* | 1 | -0.34 (0, -297.54) | 0.72 (30.89) | 0.51 (11.44) |
| BK16_B8_05 | Barkedji | Adult | Hybrid | $F_1$ | *S. bovis* | 0 | -0.34 (0, -295.66) | 0.71 (30.94) | 0.57 (11.97) |
| BK16_B8_06 | Barkedji | Adult | Hybrid | $F_1$ | *S. bovis* | 0 | -0.34 (0, -285.97) | 0.70 (29.06) | 0.56 (11.63) |
| BK16_B8_07 | Barkedji | Miracidia | Hybrid | $F_1$ | *S. bovis* | 0 | -0.34 (0, -296.37) | 0.73 (31.54) | 0.53 (12.33) |
| BK16_B8_08 | Barkedji | Miracidia | *S. curassoni* | *S. curassoni* | *S. curassoni* | 4 | -0.07 (0, -19.02) | 0.88 (60.43) | -0.02 (0.69) |
| BK16_B8_09 | Barkedji | Miracidia | *S. curassoni* | *S. curassoni* | *S. curassoni* | 0 | -0.01 (0, -2.11) | 0.88 (61.91) | -0.04 (1.42) |
| BK16_B8_10 | Barkedji | Miracidia | Hybrid | BC1 (*S. curassoni*) | *S. bovis* | 7 | -0.29 (0, -149.95) | 0.81 (44.53) | 0.45 (6.40) |
| BK16_B8_11 | Barkedji | Miracidia | *S. bovis* | BC2-BC4 (*S. bovis*) | *S. bovis* | 18 | -0.17 (0, -49.79) | 0.65 (8.54) | 0.60 (9.62) |
| BK16_B8_12 | Barkedji | Miracidia | Hybrid | *S. curassoni* | *S. curassoni* | 2 | 0 (0, 0.22) | 0.88 (65.13) | -0.05 (1.94) |
| BK16_B8_13 | Barkedji | Miracidia | *S. curassoni* | BC1 (*S. curassoni*) | *S. curassoni* | 7 | -0.31 (0, -167.74) | 0.78 (28.77) | 0.51 (6.98) |
| BK16_B8_14 | Barkedji | Miracidia | Hybrid | *S. curassoni* | *S. curassoni* | 1 | 0.02 (0, 3.71) | 0.88 (64.88) | -0.02 (0.80) |
| BK16_B8_15 | Barkedji | Miracidia | *S. curassoni* | *S. curassoni* | *S. curassoni* | 0 | 0 (0, 0.32) | 0.88 (67.53) | -0.04 (2.22) |
| BK16_B8_17 | Barkedji | Miracidia | *S. curassoni* | *S. curassoni* | *S. curassoni* | 0 | 0 (0, 0.46) | 0.88 (61.20) | -0.02 (0.61) |
| BK16_B8_19 | Barkedji | Miracidia | *S. curassoni* | *S. curassoni* | *S. curassoni* | 1 | 0.05 (0.01, 10.41) | 0.89 (62.47) | 0.04 (1.31) |
| RT15_B3_01 | Richard Toll | Adult | Unknown | *S. bovis* | *S. bovis* | 0 | -0.02 (0, -8.57) | 0.01 (0.48) | 0.65 (14.45) |
| RT15_B6_01 | Richard Toll | Adult | *S. bovis* | *S. bovis* | *S. bovis* | 0 | -0.04 (0, -15.79) | 0.05 (2.77) | 0.65 (14.20) |

We used the $f_3$ statistic to identify samples derived from admixture (Indv) between two source populations (*S. bovis*: Sb, *S. curassoni*: Sc) and the Patterson's D statistic (*D*) to differentiate between gene flow and incomplete lineage sorting (using *S. haematobium*: Sh as an outgroup population). For the $f_3$ statistic, values are given as the mean of sliding windows of 100 variants, the estimated standard error (SE) and the Z-score (Z) number of standard errors from 0. For the Patterson's D statistic, values are given as the estimated Patterson's D values and the Z-score. Backcrossed hybrids are indicated with 'BC' and the estimated number of backcrossed generations, with the backcrossed parental species in parentheses.

After an initial hybridization event, meiotic recombination over subsequent generations would be expected to break down the initial parental chromosome copies, producing a mosaic

of genomic blocks of different ancestry. To visualize the junctions in genomic ancestry that result from this recombination between heterogenic chromosomes, we repeated our ADMIX-TURE analysis in 50 kb sliding windows across each chromosome (Fig 4). Consistent with our other analyses, seven samples contained nearly complete *S. curassoni* ancestry and one sample contained nearly complete *S. bovis* ancestry. Seven samples showed consistent levels of intermediate admixture across each autosome (approximately equal *S. bovis* and *S. curassoni* ancestry). Genome-wide consistent and equal admixture between two parental populations suggests that these seven samples are first-generation hybrids between *S. curassoni* and *S. bovis*. Three of these samples showed single species ancestry assignment in the Z-specific region of the Z chromosome (found between 9.21–44.2 Mb). Males are the homogametic sex in schistosomes, so this region is not expected to recombine in females (ZW), and so would inherit this Z-specific region solely from their male parent. These three samples showed uniformly *S. curassoni* ancestry in this region, consistent with the mitotype assignment indicating *S. bovis* maternal ancestry.

The remaining three samples showed inconsistent ancestry assignment across each chromosome with large blocks of either pure or admixed ancestry. For two samples (BK16_B8_10 and BK16_B8_13), these blocks were either pure *S. curassoni* ancestry or admixed ancestry, while a third (BK16_B8_11) contained blocks of pure *S. bovis* ancestry, pure *S. curassoni* ancestry or admixed ancestry. Given the length of the assigned ancestry tracts it is likely that these samples are the results of a recent admixture event. We used the Bayesian clustering software NEWHYBRIDS [56] to assign each admixed sample to a specific generation (S10 Table). All fourteen non-admixed samples were assigned as parental species' (ten *S. bovis*, four *S. curassoni*). Consistent with our other analyses, seven samples were categorized as $F_1$. Two samples (BK16_B8_10 and BK16_B8_13) were classified as backcrosses between $F_1$ hybrids and *S. curassoni* (BC1). Both these samples also contained seven autosomal ancestry junctions consistent with a single generation since admixture (S12–S14 Figs). The remaining sample (BK16_B8_11) had an inconsistent assignment probability across NEWHYBRIDS replicates and contained 18 autosomal ancestry junctions suggesting between three and four generations since admixture (S13–S15 Figs). This would suggest several generations of backcrossing to *S. bovis* (BC2-BC4), although we could not rule out more complex hybridization scenarios.

When we compared concordance of whole-genome sequencing to the *cox1* and ITS-rDNA genotyping, we found that seven samples were correctly identified as *S. curassoni*, one sample was correctly identified as *S. bovis* and all seven $F_1$ hybrids were correctly identified as hybrid samples (S11 Table). One $F_1$ backcross (BC1) was correctly identified as being a hybrid sample but the remaining $F_1$ backcross and BK16_B8_11 did not have discordant nuclear and mitochondrial genomes and so were misidentified as *S. curassoni* and *S. bovis*, respectively. A further two *S. curassoni* samples were misidentified as admixed samples and the remaining sample could not be genotyped prior to sequencing.

## Discussion

Hybridization of schistosomes is an emerging One Health concern with the potential to have major impacts on the epidemiology and control of schistosomiasis. Therefore, it is important to identify and characterize incidence and transmission of admixed *Schistosoma* species. To investigate the genomic composition of livestock schistosomes, we performed whole-genome sequencing of individual miracidia and adult stage isolates from endemic regions, where admixed isolates are common. To support the population genomic analyses, we produced a chromosomal-scale assembly of the *S. curassoni* genome that we used to analyse population structure and ancestry of each sequenced isolate.

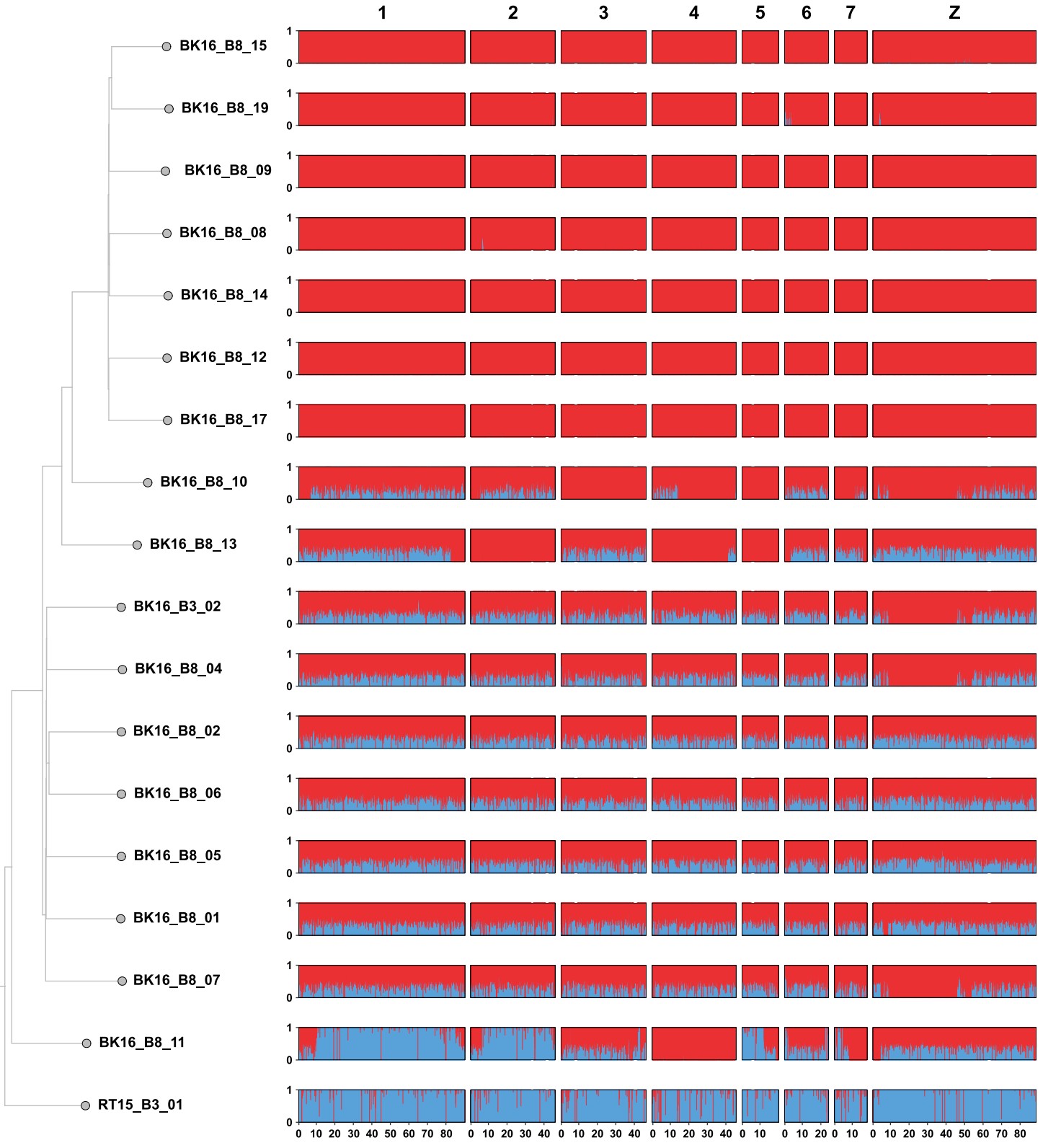

**Position on chromosome (Mb)**

**Fig 4. Per-sample estimates of ancestry.** Ancestry was estimated using ADMIXTURE in 150 kb nonoverlapping windows along each chromosome. The proportion of *S. curassoni* ancestry is shown in red, *S. bovis* in blue. Ancestry estimates are shown in phylogenetic order based on a neighbour-joining phylogeny inferred using identity-by-descent distances showing the relatedness between all samples. The scale represents the mean numbers of substitutions. Excluded from this plot are the six samples used to designate reference alleles for each population (SCUR_01, SBOV_01, SBOV_02, BK16_B3_01, BK16_B8_03, RT15_B6_01).

We successfully characterized 21 isolates sampled from four individual cattle hosts, including both miracidia and adult stage parasites from the same host. We have shown that these are most likely early generation hybrids (predominantly $F_1$ and $F_1$ backcrosses) between *S. bovis* and *S. curassoni* and we found examples of bidirectional hybridization (backcrosses to either parent). This was further supported by the mosaic composition of the backcrossed samples which showed wide variations of $f_3$, Patterson's D and ADMIXTURE estimates across the genome suggesting alternating genomic ancestry from either parent. The presence of $F_1$ and a later generation backcross suggests that these can produce viable offspring with both parental species as well as other $F_1$ hybrids.

By sampling from livestock, we were able to sequence parasites from multiple developmental stages. This is in contrast to studies sampling humans, which are typically unable to sample adult-stage worms and so are reliant only on miracidia stage parasites. While genomic measures of relatedness are unlikely to be accurate when including first- or second-generation hybrid samples, the presence of a range of adult and miracidial stage early generation hybrids from different hosts suggests that these hybrid lineages were not derived from a single hybridization event but rather multiple independent interspecies pairings.

While this study did not aim assess the overall prevalence of *S. curassoni*–*S. bovis* hybrids, the presence of multiple unrelated admixed offspring within a single host (BK16_B8) suggests that in mixed-species coinfections these two species of schistosome can hybridize. This is not surprising given the relatively recent divergence time between *S. bovis* and *S. curassoni* when compared to other species that can hybridize such as *S. bovis* and *S. haematobium* [57,58]. It is therefore likely that pre-zygotic barriers, possibly the limited overlap of the geographical distributions of these two species (and their intermediate host snails in particular), have maintained the reproductive isolation of these species. Indeed, this is consistent with genotyping surveys conducted between 2015 and 2017 in Senegal which found only limited overlap between the geographical and host ranges of these two *Schistosoma* species, at least in part restricted to their differing intermediate host snail species distribution [31]. Extensive surveys conducted in Barkedji estimated an 8% prevalence of *S. curassoni* in cattle, 84% in goats and 73% in sheep compared to a 2% prevalence of *S. bovis* across all three host types [31]. In contrast, in Richard Toll and the Lac de Guiers the same authors estimated a 1% prevalence of *S. curassoni* in cattle and sheep and 11% in goats but a 92% prevalence of *S. bovis* among cattle, a 14% prevalence in sheep and 15% prevalence in goats. Whilst any current/previous geographical barriers would have assisted with the reproductive isolation of these species this also means, however, that changes in human or livestock migration patterns or differential control pressures have the potential to radically alter the prevalence of hybrid and non-hybrid schistosomes [59]. This may be particularly pertinent given the evidence that hybrid *Schistosoma* can also expand their molluscan intermediate host range [25,28], and that schistosomes displaying markers from two livestock-infective schistosome species have been identified to infect humans [30]. Furthermore, while our whole-genome sequencing results were largely consistent with the widely-used dual marker approaches (such as cox1/ITS-rDNA genotyping), we found examples of gains in sensitivity suggesting that the prevalence of hybrid lineages has likely been under-estimated previously. If our observations of novel *S. curassoni*–*S.bovis* hybrid lineages are

representative of livestock populations in general, then it appears that new allelic combinations are continuously being introduced into schistosome populations.

The long-term survival of these hybrid lineages will be determined, at least in part, by the prevalence of non-hybrid lineages. In Barkedji, where all admixed samples were found, the estimated prevalence of *S. curassoni*–*S. bovis* hybrids across all three host types varied between 2–4% [31]. In the same study, the authors did not find snails shedding *S. bovis* in Barkedji concluding that *S. bovis* was most likely imported from neighbouring regions. Based on the overall prevalence estimates of *S. curassoni*, *S. bovis* and hybrids in Barkedji we would expect the majority of hybrid lineages to not be maintained in the population and instead repeatedly backcross with *S. curassoni* [59].

However, multiple studies have observed that hybrid lineages from closely-related species display increased fitness compared to non-hybrid lineages. For example, within mixed host infrapopulations, *S. guineensis* and *S. haematobium* females will preferentially form heterospecific pairings with *S. guineensis*–*S. haematobium* hybrid males compared to homospecific pairings to males of their own species [60,61]. These and other competitive mating interactions have been suggested as a factor in the declining prevalence of *S. guineensis* in Cameroon [62,63]. Conversely, if the *S. bovis*-*S. curassoni* hybrids display reduced fitness, as observed in pairings between distantly-related schistosome species [64–66], then they would be less likely to be maintained in the population. This ambiguity underscores the importance of epidemiological surveys to determine to what extent novel hybrid lineages are spreading through livestock populations.

To conclude, the threat to public and animal health that is presented by parasite hybridization and zoonotic spillover is predicted to increase with rapid anthropogenic changes and global trends such as migration and changing land and agricultural practices. Our study, using WGS in combination with *de novo* genome assembly, provides conclusive evidence of contemporary hybridization between schistosomes species under natural settings. We found multiple unrelated generations of hybrids within the same definitive host demonstrating that these hybrids are viable beyond the $F_1$ generation. We now recommend more extensive, longitudinal genomic surveys to determine the long-term viability of these hybrids and to search for evidence of adaptive introgression from historic hybridization events. Our *de novo* genome assembly of *S. curassoni* presented here will be a valuable resource for future investigations into both livestock schistosomes and hybrid infections of both humans and animals alike. Such studies should provide greater insights into the underlying mechanisms maintaining species boundaries and the long-term genomic impacts of hybridization in general. Given the potential for zoonotic risk of livestock hybrid schistosomes our results also highlight the importance of considering control measures in a One Health framework. Understanding how these parasite populations are adapting to recent anthropogenic changes will be vital to predicting the outcomes and long-term successes of current elimination efforts, as well as to forecast the likely spread of tropical diseases such as schistosomiasis beyond their original host and geographic boundaries.

## Methods

### Ethics statement

Ethical approval for sampling was provided by: (i) Imperial College (London, UK) application 03.36; (ii) the Clinical Research and Ethical Review Board at the Royal Veterinary College (London, UK) application URN20151327; (iii) the Comité National d'Ethique pour la Recherche en Santé (Dakar, Senegal) application SEN15/68. Written informed consent was obtained from all livestock owners. Livestock owners were offered standard anthelmintics for infected animals.

## Schistosoma curassoni genome assembly

A pool of 20 adult *Schistosoma curassoni* males was obtained from the Schistosome Collection at the Natural History Museum (SCAN) [67]. These represent the second passage of field isolates originally sampled from Dakar, Senegal and were stored in liquid nitrogen on 4th August 1993. DNA was extracted using a modified agarose plug based protocol (Bionano Prep Animal Tissue DNA Isolation Soft Tissue Protocol, Bionano Genomics, San Diego, CA, USA) yielding a total of 2.6 μg of high-molecular weight DNA. The eluted DNA was cleaned using standard phenol/chloroform extraction protocols, DNA concentrations were determined using Qubit high sensitivity kit and DNA size distributions were confirmed using FEMTO Pulse (Agilent, Santa Clara, CA, USA). A Pacific Bioscience CLR long-read sequencing library was constructed using the SMRTbell Template Prep Kit 1.0 and sequenced on the Sequel II platform, by the Scientific Operations core at the Wellcome Sanger Institute. This yielded a total of 38.61 Gb of PacBio subreads (N50 = 16.6 kb, S12 Table).

PacBio subreads were assembled using the FALCON assembly and FALCON-unzip (falcon kit v1.8.1). For initial quality control, primary contigs and alternate haplotigs were merged into a single assembly. PacBio subreads were mapped to the assembly using Minimap2 (v2.16-r922) [68]. Taxonomic annotation of contigs was performed using both Blastn (v.2.2.31) and Diamond (v0.9.14.115) against NCBI nt (v2.10.1+) and UniProt Reference Proteomes (Release 2017_07) databases respectively. Using the mapping and taxonomic annotation files Blobtools was used to remove contigs representing potentially contaminants (total = 69.61 kb) [69] (S16 Fig). Two rounds of error correction were performed by first mapping PacBio subreads to the genome using pbmm2 (v1.4.0) and then error correction was performed using gcpp (v2.0.0). Alternate haplotigs identified using FALCON-Unzip were first removed and any remaining alternate haplotigs were identified using purge_dups [70]. Arima Hi-C data (S12 Table) was then used to scaffold the primary assembly into chromosome spanning scaffolds using the Juicer and 3D-DNA pipelines [71,72]. PacBio pre-assembled reads (from FALCON assembly) were mapped to the genome using minimap2 and the assembly was visualised using Gap5 [73]. Hi-C reads were mapped to the assembly using bwa mem and visualised using PretextView. The assembly was then manually curated in Gap5 using the Hi-C data as a guide to break misjoins and scaffold over repetitive or highly haplotypic regions. Then a final round of polishing using pbmm2 and GCPP was performed as above.

We assessed the completeness of the curated assembly by searching for conserved, single copy genes using BUSCO (Benchmarking Universal Single-Copy Orthologs) (v3.0.2) and the eukaryota lineage (odb10) dataset [74]. Juicer was used to map all Hi-C data to the final assembly and Juicebox was used to visualise the data. Pseudoautosomal regions (PARs) of the Z chromosome were identified based on synteny to *S. mansoni*. Annotation of repetitive elements was conducted using RepeatModeler and RepeatMasker [75]. Per chromosome GC content was measured using Bedtools [76], assembly features were visualised using the circlize package [77].

## Genome annotation

Gene structures of conserved *S. mansoni* genes were predicted using GenomeThreader (v1.7.1) [78], using spliced alignments of *S. mansoni* (v9) transcript and protein sequences [47].

## Synteny inference

Comparison of synteny between *S. curassoni* and *S. mansoni* were performed using MCscan [79]. Gene coordinates were extracted from GFF3 files and pairwise synteny was inferred

using coding sequences, setting the cscore cutoff to 0.99. Macrosynteny was visualised using the JCVI utility libraries [80].

## Sampling of abattoir field isolates

Post-mortem abattoir surveys of routinely slaughtered cattle, sheep and goats were carried out between 2015 and 2017 in three villages of Northern Senegal, Richard Toll (16.47057, -15.694457) in the Senegal River Basin and Linguère (15.24241, -15.06592) and Barkedji (15.28243333, -14.87238333) in the Vallée du Ferlo [31]. In Richard Toll water bodies are permanent and transmission is perennial. Due to the proximity of the Diama Dam, the area has undergone substantial land-use changes, with desalination and creation of irrigation canals that have facilitated expansion of habitats for snail intermediate hosts and increased sharing of water contact points by communities and their animals. The main livestock-schistosome species circulating in this area is *S. bovis*. By contrast, in Barkedji and Linguère, temporary ponds are the main water sources. Schistosomiasis transmission is seasonal as they disappear completely during the dry season which necessitates seasonal migration by a large proportion of livestock-keeping communities. The main livestock-schistosome species circulating in this area is *S. curassoni* [31]. During post-mortem abattoir surveys of routinely slaughtered animals, mesenteric and rectal blood vessels and bladder and associated vasculature were visually inspected for the presence of adult-stage worms. Fecal samples were collected directly from the rectum as well sections of liver, spleen, lungs and kidney and processed following an adapted miracidial hatching technique [31]. Urine was collected for filtration. All adult-stage worms (single males, females and split couples) were stored in RNA-later. Following urine filtration and miracidia hatching of faeces and tissue samples, free-swimming miracidia were individually pipetted onto Whatman Indicating FTA Classic cards (GE Healthcare Life Sciences, UK).

## Genotyping field isolates

Single miracidia were isolated from Whatman Indicating FTA Classic cards using a Harris micro-punch. DNA was extracted from single miracidia using the CGP protocol originally described by Moore et al. [81] and used to sequence *S. mansoni* miracidia by Doyle et al. [82].

Adult-stage worm samples were manually inspected and separated so only individuals were sequenced. DNA was extracted from these samples using the QIAGEN DNeasy Blood & Tissue Kit (QIAGEN, Hilden, Germany), following manufacturer's instructions.

Partial fragments of the mitochondrial cytochrome c oxidase subunit 1 (*cox1*) and the complete nuclear ribosomal DNA internal transcribed spacer (ITS) were amplified from individual DNA extracts [83,84](S13 Table). The following PCR conditions were used: initial denaturation step of 5 minutes at 95˚C, 40 cycles of 30 seconds at 95˚C, 30 sec at 58˚C and 1 min at 72˚C; final extension period of 7 min at 72˚C. Polymerase chain reaction (PCR) fragments were sequenced by Eurofins Genomics (Cologne, Germany) using original primers. Codon-Code Aligner v7.0.1 (Centerville, USA) was used to manually edit and assemble sequences and were compared to *Schistosoma* reference sequences to confirm species. Adult-stage worms or miracidia displaying both *S. bovis* and *S. curassoni* signals were designated as hybrids.

## Sequencing of field isolates

DNA sequencing libraries for all miracidia samples were prepared using the laser capture microdissected biopsy (LCMB) protocol described by Lee-Six et al. [85]. In short, 20 μl of lysate was mixed with 50 μl of Ampure XP beads and 50 μl of TE buffer (Ambion 10 mM Tris-HCL, 1 mM EDTA), followed by incubation (5 minutes, room temperature). Tubes were placed on magnetic racks and beads were washed twice with 75% ethanol, followed by

resuspension in 26 μl of TE buffer. DNA fragmentation and A-tailing was performed by mixing each sample with 7 μl of 5x Ultra II FS buffer and 2 μl of Ultra II FS enzyme (New England BioLabs) followed by two incubation steps (12 minutes at 37˚C then 30 minutes at 65˚C). For adaptor ligation, 1 μl ligation enhancer (New England BioLabs), 0.9 μl nuclease-free water (Ambion), 30 μl of ligation mix and 0.1 μl duplexed adapters were added to each sample followed by incubation (20 minutes, 20˚C). Library purification and elution was performed by adding 65 μl of Ampure XP beads and 65 μl of TE buffer. 25 μl KAPA HiFi HotStart ReadyMix (KAPA Biosystems) and 1 μl PE1.0 primer to 21.5 μl of eluted library and each sample was thermal-cycled as follows: 98˚C for 5 mins, then 12 cycles of 98˚C for 30 secs, 65˚C for 30 sec, 72˚C for 1 min and 72˚C for 5 mins. Amplified libraries were purified using a 0.7:1 volumetric ratio of Ampure beads to library, followed by elution in 25 μl of nuclease-free water. Multiplexed 150 bp paired-end read libraries were sequenced using the Illumina Hiseq X system at the Wellcome Sanger Institute. Cluster generation and sequencing were undertaken according to the manufacturer's protocol.

For all adult-stage samples DNA was quantified with the Biotium Accuclear Ultra high sensitivity dsDNA Quantitative kit using Mosquito LV liquid platform, Bravo WS and BMG FLUOstar Omega plate reader. For each sample, 200ng/120μl was used to prepare sequencing libraries. DNA was sheared to target 450 bp fragment size using a Covaris LE220 instrument, samples were then purified using Agencourt AMPure XP SPRI beads using the Agilent Bravo WS automation system. Library construction was performed using a NEB Ultra II DNA library preparation kit using the Agilent Bravo WS automation system. PCR was performed using the KapaHiFi Hot start mix and IDT 96 iPCR tag barcodes. 6 standard cycles of PCR were performed using the following conditions: initial denaturation step of 5 minutes at 95˚C, 30 seconds at 98˚C, 30 sec at 65˚C and 1 min at 72˚C; final extension period of 10 min at 72˚C. Libraries were then purified using Agencourt AMPure XP SPRI beads on Beckman BioMek NX96 liquid handling platform and libraries were quantified with the Biotium Accuclear Ultra high sensitivity dsDNA Quantitative kit using Mosquito LV liquid platform, Bravo WS and BMG FLUOstar Omega plate reader. Libraries were pooled in equimolar amounts and normalised to 2.8nM. Multiplexed 150 bp paired-end read libraries were sequenced using the Illumina Hiseq X10 system at the Wellcome Sanger Institute. Cluster generation and sequencing were undertaken according to the manufacturer's protocol.

## Reference samples

We downloaded additional genome sequence data from the NCBI Short Read Archives for seven other species in the Haematobium clade as reference samples for each species. This included read data for each of the following species: *S. curassoni*, *S. guineensis* (unpublished), *S. intercalatum* (unpublished), *S. matthei* and *S. margrebowiei* [48] two *S. bovis* [48,86] and *S. haematobium* [44].

Sequence data for *S. guineensis* and *S. intercalatum* were originally sequenced as part of a consortia project [48] but were not included in the publication, below we have detailed the sample background and sequencing methods as background for future work. (S14 Table). *S. guineensis* was collected in Chacara, Sao Tomé and *S. intercalatum* from the Democratic Republic of Congo. Both were passaged through mice at the Natural History Museum, London. DNA was extracted from a single adult-stage male worm of each isolate by phenol-chloroform extraction, sheared to target 450bp fragment size using a Covaris LE220 instrument, purified with AMPure XP SPRI beads and sequencing libraries generated using the NEB Ultra II library prep kit and 8 cycles of PCR before sequencing 200bp paired-end reads on a HiSeq 2000 platform.

## Sequence analysis

Sequence reads were aligned to the *S. curassoni* reference assembly (all autosomes, Z chromosome and the mitochondrial genome) using BWA mem (v.0.7.17) and duplicates were marked using PicardTools MarkDuplicates (as part of GATK v.4.1.0.0). Per-sample variant calling was performed using GATK HaplotypeCaller and consolidated using GATK CombineGVCFs. Joint-call cohort genotyping was performed using GATK GenotypeGVCFs. We partitioned and filtered SNPs and indels/mixed (variant sites that had both SNPs and indels called) sites separately using GATK SelectVariants and filtered using VariationFiltration, respectively. SNPs were retained if they met the following criteria: QD $\geq$ 2.0, FS $\leq$ 60.0, MQ $\geq$ 40.0, MQRankSum $\geq$ -12.5, ReadPosRankSum $\geq$ -8.0, SOR $\leq$ 3.0 (S17 Fig). Indels and mixed sites were filtered independently and variants were retained if they met the following criteria: QD $\geq$ 2.0, FS $\leq$ 200.0, ReadPosRankSum $\geq$ -20.0, SOR $\leq$ 10.0 (S17 Fig). We removed all sites with high proportions of SNP missingness (>5%) using BCFtools [87]. For the rest of the genome, we first excluded a single sample (BK16_B8_18) due to high rates of SNP missingness (>90% of all variant sites could not be genotyped). We then removed sites with a high proportion of SNP missingness across samples (>5%) using BCFtools. We called variants against the mitochondrial genome using GATK HaplotypeCaller in haploid mode but otherwise followed the same filtering parameters as the non-mitochondrial variants and excluded the BK16_B8_18 as above.

We identified samples with same-species contamination by examining allelic imbalances in heterozygous variants, where the ratio of reads supporting each allele substantially deviates from 50% (based on methods described by https://speciationgenomics.github.io/allelicBalance/) (S5–S8 Figs). Results were manually inspected and non-reference samples with clear, continuous peaks of allelic ratios at >0.6 or <0.4, were classified as contaminated. This identified three samples with evidence of contamination and we excluded them from further analysis. As additional lines of evidence, for each sample we estimated the fraction of reads coming from cross-sample contamination using GetPileupSummaries and CalculateContamination (as implemented in GATK v.4.2.0.0; S4 Table). Finally, for all samples not removed due to contamination, we estimated the average number of haplotypes in each sample using estMOI (v.1.03; S5 Table) [88].

## Depth of coverage

We calculated coverage in 25 kb windows along each chromosome using bedtools coverage. We calculated median read coverage across the pseudoautosomal regions (PARs) of the Z chromosome: PAR1 (coordinates 0–9,212,000 bp), PAR2 (coordinates 44,200,000–88,179,078 bp) and the non-pseudoautosomal: Z-specific region (ZSR, coordinates 9,212,000–44,200,000 bp). To differentiate male and female samples we compared the relative coverage over the ZSR and PAR regions. Samples with a PAR/ZSR ratio >0.75 were classified as male and PAR/ZSR ratio < 0.75 were classified as female.

## Population structure

We used PLINK (v2.0) to remove variants in strong linkage disequilibrium (LD), discarding variants according to the observed correlation between pairs of variant sites. We scanned the genome in sliding windows of 50 variants, in steps of 10 variants and removed variants with squared correlation coefficients > 0.15. We also excluded all non-autosomal variants (those found on the Z-chromosome and mitochondrial genome) resulting in 366,486 autosomal variants. Principal component analysis (PCA) was performed using PLINK. We used publicly available scripts to convert 366,486 autosomal SNPs into Phylip format (github.com/edgardomortiz/vcf2phylip/vcf2phylip.py) followed by removal of all invariant sites (github.com/

btmartin721/raxml_ascbias/ascbias.py). IQ-TREE was used to perform phylogenetic inference using the best-fit substitution model with ascertainment bias correction selected by ModelFinder (TVM+F+ASC+R2) and 1000 ultrafast bootstraps. The resulting phylogeny was visualised using ggtree (v.1.10.5) [89]. We then repeated these analyses for the mitochondrial variants without LD pruning. Additionally, we used PLINK (v1.9) to produce an identity-by-state distance matrix from a subset of samples and autosomal variants. We then used ape (v5.2) bionj algorithm to produce a neighbour-joining phylogeny. The resulting phylogeny was visualised using ggtree.

## Ancestry estimation

Using the same LD pruned variant set as the previous section was converted into a BED file using PLINK (2.0) and we then estimated individual ancestry using ADMIXTURE [90] including only *S. curassoni*, *S. bovis* and *S. haematobium* samples. We used a range of K values (number of hypothetical ancestral populations) ranging from 1–20, 10-fold cross-validation, standard error estimation with 250 bootstraps and repeated the process 10 times with different random seeds.

We then used PLINK to divide variants into 50 kb non-overlapping windows across each autosome and the Z chromosome. We designated reference populations for both *S. curassoni* (SCUR_01, BK16_B3_01 and BK16_B8_03) and *S. bovis* (SBOV_01, SBOV_02, RT15_B6_01) selected from published datasets or those with no evidence of admixture (based on the ADMIXTURE results and *cox1*/ITS-rDNA). We repeated the ADMIXTURE analysis across each 50 kb window without bootstrapping, specifying two populations (K = 2) and using the six reference samples as designated populations for each species. We produced a second autosomal neighbour-joining phylogeny (as above) excluding six samples (SCUR_01, SBOV_01, SBOV_02, BK16_B3_01, BK16_B8_03, RT15_B6_01) and plotted ancestry estimates in phylogenetic order.

Autosomal junctions (switches in genomic ancestry resulting from recombination between heterogenic chromosomes) were identified using ADMIXTURE. We created 1 Mb sliding windows along each chromosome, proceeding in steps of 500 kb, and calculated the ADMIXTURE value for each window as above. We created 3 ancestry categories: predominantly *S. bovis* ancestry (*S. bovis* admixture proportion > = 0.75), predominantly *S. curassoni* ancestry (*S. bovis* admixture proportion < = 0.25), admixed (*S. bovis* admixture proportion >0.25 & <0.75). We defined an ancestry junction as any region of the genome where one or more 500 kb windows were in a different category than the previous window. We estimated the expected number of autosomal junctions for each generation since an initial hybridization event using the junctions R package [91]. We tested values of initial heterozygosity, assuming an infinite population size and an autosomal genetic map length of 983.9 cM.

We used scikit-allel to calculate the average outgroup $f_3$ statistic for each sample, standard errors were estimated with a block jackknife over 1000 markers (blen = 1000). We defined three populations described by the tree ((P1,P2),P3). We designated reference populations for *S. bovis* (SBOV_01, SBOV_02, RT15_B6_01) and *S. curassoni* (SCUR_01, BK16_B3_01 and BK16_B8_03) as P1 and P2, respectively. Individual samples to be analysed were designated at P3. We used Dsuite (v0.5 r45) to calculate Patterson's D (ABBA-BABA) statistics using all autosomes [92]. The test was performed using four populations P1, P2, P3, and O, described by the rooted tree (((P1,P2),P3),O). *S. haematobium* was designated as the outgroup population (O), samples to be analysed (newly sequenced non-reference samples) were designated P2, reference populations for *S. curassoni* and *S. bovis* were designated as either P1 or P3 (analysis was performed with populations in each position).

The Bayesian clustering software NEWHYBRIDS was used to assign each sample into different hybrid categories (pure *S. bovis*, pure *S. curassoni*, $F_1$, $F_1$ backcross or $F_2$). NEWHYBRIDS can only be run efficiently on a small number of loci, so we randomly subsampled 300 bi-allelic autosomal variants and used PGDSpider to convert the subset VCF file into NEWHYBRIDS format. NEWHYBRIDS was run for 150,000 sweeps of burn-in and 500,000 sweeps of data collection. We repeated this analysis 14 additional times using different random subsets of 300 variants for each replicate.

### Relatedness estimation

We used NGSremix to estimate pairwise relatedness between all *S. bovis*, *S. curassoni* and admixed samples [93]. The LD pruned BED file was used as input along with admixture proportions and ancestral allele frequencies estimated by ADMIXTURE. The summary statistics $k_0$, $k_1$, $k_2$ (the proportion of the genome where a pair of individuals share 0,1 or 2 alleles IBD, respectively) were used to estimate relatedness. Values of $(k_1, k_2)$ of (0,0) corresponded to unrelated pairs, (1,0) for parent-offspring, (0.5,0.25) for full siblings, (0.25,0) for first-cousins and (0.625,0) for second cousins [94].

### Estimating fixed variants between species

We identified biallelic SNP sites found as homozygous reference alleles in three non-admixed *S. curassoni* samples (SCUR_01, BK16_B3_01 and BK16_B8_03) and as homozygous alternate alleles three non-admixed *S. bovis* samples (SBOV_01, SBOV_02, RT15_B6_01). This identified 309,040 SNPs fixed between these two populations, per SNP site statistics were calculated for the remaining samples using BCFtools.

### Supporting information

**S1 Table. Sample metadata.**
(CSV)

**S2 Table. Depth of read coverage statistics.** For each sample the columns show total number of reads, the proportion of reads mapping to the Schistosoma curassoni genome and the ratio in coverage between the pseudoautosomal regions (PAR) and the Z-specific region (ZSR).
(CSV)

**S3 Table. Depth of read coverage statistics per-chromosome.** Per-sample depth of read coverage across each chromosome.
(CSV)

**S4 Table. Estimates of cross-individual contamination using CalculateContamination (as implemented in GATK v.4.2.0.0).**
(CSV)

**S5 Table. Haplotype estimation based contamination screening.** We used estMOI to estimate the number of haplotypes per sample, i.e. the number of distinct haplotypes; reported as the multiplicity of infection (MOI) estimates.
(CSV)

**S6 Table. Variant summary statistics.** Per-sample summary statistics of post-quality control variant calls as classified by BCFtools.
(CSV)

**S7 Table. Ancestry estimation.** Per-sample ADMIXTURE results assuming two populations (K = 2).
(CSV)

**S8 Table. Sample relatedness.** Estimation of relatedness between pairs of samples using NGSremix. k0, k1, and k2 indicate the proportion of each genome where each pair of individuals share 0, 1, or 2 alleles identical-by-descent, respectively.
(CSV)

**S9 Table. Identification of fixed alleles in S. bovis and S. curassoni poplations.** Variants found as homozygous reference alleles in all three indicated *S. curassoni* samples and as homozygous alternate alleles in all three indicated *S. bovis* populations were considered potentially fixed variants between species.
(CSV)

**S10 Table. NEWHYBRIDS assignment probability.** The Bayesian clustering software NEW-HYBRIDS was used to assign each sample into different hybrid categories (pure *S. bovis*, pure *S. curassoni*, F1, F2 or F1 backcross). We randomly subsampled 15 sets of 300 bi-allelic autosomal variants and ran NEWHYBRIDS for 150,000 sweeps of burn-in and 500,000 sweeps of data collection. Results are reported as the mean and standard deviation (in parentheses) of all 15 runs.
(CSV)

**S11 Table. Comparison between single-locus genotyping and whole-genome sequencing species identification results.**
(CSV)

**S12 Table. PacBio and Arima Hi-C sequence metadata.**
(CSV)

**S13 Table. PCR primers used for species identification.**
(CSV)

**S14 Table. Read sequence metadata.** Raw Illumina reads used for population genomic analyses, location of reads.
(CSV)

**S1 Fig. Genome-wide sequence coverage for each reference sample.** Illumina reads were mapped to the chromosomes of our *S. curassoni* reference assembly. Median read depth calculated for 25 kb windows along each chromosome.
(EPS)

**S2 Fig. Genome-wide sequence coverage for each adult-stage parasite from host BK16_B8.** Illumina reads were mapped to the chromosomes of our *S. curassoni* reference assembly. Median read depth calculated for 25 kb windows along each chromosome.
(EPS)

**S3 Fig. Genome-wide sequence coverage for each miracidial-stage parasite from host BK16_B8.** Illumina reads were mapped to the chromosomes of our *S. curassoni* reference assembly. Median read depth calculated for 25 kb windows along each chromosome.
(EPS)

**S4 Fig. Genome-wide sequence coverage for each adult-stage parasite from all remaining adult stage parasites (hosts RT15_B3, RT15_B6 and BK16_B3).** Illumina reads were mapped to the chromosomes of our *S. curassoni* reference assembly. Median read depth

calculated for 25 kb windows along each chromosome.
(EPS)

**S5 Fig. Contamination screening of reference samples.** The first column shows the number of reads supporting each allele at every heterozygous genotype for each sample. Black lines show the expected distribution of a 50:50 allelic ratio (upper) and a 60:40 allelic ratio (lower). The second column shows the proportion of minor allele reads supporting each heterozygous genotype.
(EPS)

**S6 Fig. Contamination screening of adult stage parasites from host BK16_B8.** The first column shows the number of reads supporting each allele at every heterozygous genotype for each sample. Black lines show the expected distribution of a 50:50 allelic ratio (upper) and a 60:40 allelic ratio (lower). The second column shows the proportion of minor allele reads supporting each heterozygous genotype.
(EPS)

**S7 Fig. Contamination screening of miracidial stage parasites from host BK16_B8.** The first column shows the number of reads supporting each allele at every heterozygous genotype for each sample. Black lines show the expected distribution of a 50:50 allelic ratio (upper) and a 60:40 allelic ratio (lower). The second column shows the proportion of minor allele reads supporting each heterozygous genotype.
(EPS)

**S8 Fig. Contamination screening of all remaining adult stage parasites (hosts RT15_B3, RT15_B6 and BK16_B3).** The first column shows the number of reads supporting each allele at every heterozygous genotype for each sample. Black lines show the expected distribution of a 50:50 allelic ratio (upper) and a 60:40 allelic ratio (lower). The second column shows the proportion of minor allele reads supporting each heterozygous genotype.
(EPS)

**S9 Fig. Midpoint rooted maximum-likelihood phylogeny inferred using 23,136 parsimony informative single-nucleotide polymorphisms.** Tips show the 21 isolates sequenced for this study (grey circles) and the 12 samples included as references for each available Haematobium clade species (triangles): *S. margrebowiei* (yellow), *S. mattheei* (pink), *S. intercalatum* (orange), *S. guineensis* (green) *S. haematobium* (purple), *S. curassoni* (red) and *S. bovis* (blue). Node labels represent support values from 1000 ultrafast bootstrap replicates.
(EPS)

**S10 Fig. Coefficient of variation values generated by ADMIXTURE with K values ranging from 1–20, 10-fold cross-validation and standard error estimation with 250 bootstraps.** The analysis was repeated 10 times with different random seeds, grey points represent each of the 10 measurements.
(EPS)

**S11 Fig. Population structure of sequenced isolates using mitochondrial variants.** Samples denoted with triangles represent reference samples, circles represent samples sequenced for this study. Maximum likelihood phylogeny showing the relatedness between all samples, using 581 parsimony informative sites. Plots (A) and (B) represent different visualisations of the same phylogeny. Node labels represent the bootstrap values derived from 1000 ultrafast bootstraps. Node labels in (A) represent support values from 1000 ultrafast bootstrap replicates.
(EPS)

**S12 Fig. Identification of autosomal junctions using ADMIXTURE (plot 1 of 3).** We created 1 Mb sliding windows along each chromosome, proceeding in steps of 500 kb (shown as black points). We estimated the ancestry on each window using ADMIXTURE and created 3 ancestry categories predominantly *S. bovis* ancestry (*S. bovis* admixture proportion $> = 0.75$, blue), predominantly *S. curassoni* ancestry (*S. bovis* admixture proportion $< = 0.25$, red), admixed (*S. bovis* admixture proportion $>0.25$ & $<0.75$, white). We defined an ancestry junction as any region of the genome where one or more 500 kb windows were in a different category than the previous window (indicated by a line crossing the boundaries between categories.
(EPS)

**S13 Fig. Identification of autosomal junctions using ADMIXTURE (plot 2 of 3).** We created 1 Mb sliding windows along each chromosome, proceeding in steps of 500 kb (shown as black points). We estimated the ancestry on each window using ADMIXTURE and created 3 ancestry categories predominantly *S. bovis* ancestry (*S. bovis* admixture proportion $> = 0.75$, blue), predominantly *S. curassoni* ancestry (*S. bovis* admixture proportion $< = 0.25$, red), admixed (*S. bovis* admixture proportion $>0.25$ & $<0.75$, white). We defined an ancestry junction as any region of the genome where one or more 500 kb windows were in a different category than the previous window (indicated by a line crossing the boundaries between categories.
(EPS)

**S14 Fig. Identification of autosomal junctions using ADMIXTURE (plot 3 of 3).** We created 1 Mb sliding windows along each chromosome, proceeding in steps of 500 kb (shown as black points). We estimated the ancestry on each window using ADMIXTURE and created 3 ancestry categories predominantly *S. bovis* ancestry (*S. bovis* admixture proportion $> = 0.75$, blue), predominantly *S. curassoni* ancestry (*S. bovis* admixture proportion $< = 0.25$, red), admixed (*S. bovis* admixture proportion $>0.25$ & $<0.75$, white). We defined an ancestry junction as any region of the genome where one or more 500 kb windows were in a different category than the previous window (indicated by a line crossing the boundaries between categories.
(EPS)

**S15 Fig. Estimation of the expected number of autosomal junctions per-generation since admixture.** For different initial heterozygosities we plotted the expected number of junctions per generation assuming an infinite population size and an autosomal genetic map length of 983.9 cM.
(EPS)

**S16 Fig. BlobToolKit plot of the S. curassoni assembly (schistosoma_curassoni_626247_v1.0).** The x-axis values show the GC-proportion for each chromosome and unplaced contig, y-axis values show the coverage of the PacBio subreads mapped to the assembly. Chromosome and unplaced contig sizes are shown by the size of each circle and circles are coloured by taxonomic annotation.
(PNG)

**S17 Fig. Variant quality control.** Plots A-F show the frequency distribution of annotation values for 17,395,313 single nucleotide polymorphisms (SNPs) (light blue), indels and mixed sites (SNPs and indels called at the same position for different samples) (red). Plots G-H show the frequency distribution of per-sample missingness (samples with a high rate of per-site variant missingness) and per-site (sites with a high proportion of variant missingness) missingness

after filtering using the thresholds in A-F for all remaining variants (purple). Vertical dashed lines show the thresholds applied in the study for removing sites. Samples or sites were removed if they exceeded the thresholds.
(PDF)

## Acknowledgments

We would like to thank the DNA pipeline operations and informatics teams at the Wellcome Sanger Institute for production of the sequence data. We would also like to thank Sarah Buddenborg for her helpful feedback on earlier versions of this manuscript. The authors are extremely grateful to the people who helped in the collection of the samples: Dr Anna Borlase, Mr. Samba D. Diop, Dr Stefano Catalano, Dr Cheikh Binetou Fall, Mr. Cheikh Tidiane Thiam and Mr Alassane Ndiaye.

## Author Contributions

**Conceptualization:** Elsa Léger, Joanne P. Webster.

**Data curation:** Duncan J. Berger.

**Formal analysis:** Duncan J. Berger.

**Funding acquisition:** Matthew Berriman, Joanne P. Webster.

**Investigation:** Duncan J. Berger, Elsa Léger, Geetha Sankaranarayanan, Mariama Sène, Nicolas D. Diouf, Muriel Rabone, Aidan Emery, Joanne P. Webster.

**Methodology:** Duncan J. Berger, Elsa Léger, Joanne P. Webster.

**Project administration:** Duncan J. Berger, Elsa Léger, Mariama Sène, Nicolas D. Diouf, Joanne P. Webster.

**Resources:** Muriel Rabone, Aidan Emery, Fiona Allan.

**Supervision:** Matthew Berriman, Joanne P. Webster.

**Visualization:** Duncan J. Berger.

**Writing – original draft:** Duncan J. Berger.

**Writing – review & editing:** Duncan J. Berger, Elsa Léger, James A. Cotton, Matthew Berriman, Joanne P. Webster.

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
