## [Decision Letter · Decision Letter 0]

26 Apr 2022

Dear Mr Berger,

Thank you very much for submitting your manuscript "Genomic evidence of contemporary hybridization between Schistosoma species" for consideration at PLOS Pathogens. As with all papers reviewed by the journal, your manuscript was reviewed by members of the editorial board and by several independent reviewers. The reviewers appreciated the attention to an important topic. Based on the reviews, we are likely to accept this manuscript for publication, providing that you modify the manuscript according to the review recommendations.

Sincerely,

Mostafa Zamanian, Ph.D.

Guest Editor

PLOS Pathogens

James Collins III

Section Editor

PLOS Pathogens

Kasturi Haldar

Editor-in-Chief

PLOS Pathogens

orcid.org/0000-0001-5065-158X

Michael Malim

Editor-in-Chief

PLOS Pathogens

orcid.org/0000-0002-7699-2064

Reviewer Comments (if any, and for reference):

Reviewer's Responses to Questions

**Part I - Summary**

Reviewer #1: This is a population genomic study of the parasites Schistosoma bovis and Schistosoma curassoni obtained from slaughtered livestock in Senegal. The authors generated a new genome assembly for S. curassoni and aligned whole genome sequencing reads from 21 samples. They infer that a large number of sequenced parasites are first- or second-generation hybrids, a conclusion supported by several independent analyses. They discuss these results in light of parasite population dynamics and disease control. Overall, I find their conclusions to be robust. The main conclusion of recent hybridization is supported by multiple lines of evidence. By examining such patterns in windows across the genome, it’s clear that ancestry is either consistently intermediate across the genome (F1) or choppy with large sections matching one or the other ancestral species (backcross, F2, etc.).

Reviewer #2: This paper uses WGS to investigate the ancestry of adult schistosome worms and miracidia sampled from livestock in Senegal, in order to show that hybridisation among cattle schistosomes is an ongoing process. The authors advance a new high-quality reference genome and utilise a series of high-standard population genomic analyses. Their methods/approach are appropriate and well performed to reach the aims of the study. The results are clearly presented, and the discussion is written to the point. Hence, I have no major issues with this paper and would recommend its publication.

Reviewer #3: SUMMARY

=======

This is a review of the manuscript entitled "Genomic evidence of contemporary hybridization between Schistosoma species" (PPATHOGENS-D-22-00383) by Berger, Léger et al. In their manuscript, the authors analyzed whole genomes of schistosome individuals (adult and miracdidium stage) obtained from livestock known to be infected with *Schistosoma curassoni* and *S. bovis*. This analysis aims to better understand what is the natural occurrence of hybrids in veterinary relevant schistosomes and what can be the history of hybrid lineages. The results revealed the presence of pure *S. curassoni* and *S. bovis* as well as hybrids (majority of F1s, some F1 and F2 backcrosses) mainly in a single host. These results were compared to hybrid identification using single nuclear/mitochondrial markers on the same sample which showed and expected greater sensitive of whole genome approach since some samples were misassigned as pure sample species with markers while in fact being hybrids. This study also provided a very good assembly of *S. curassoni* genome which is critical for the current study and will be valuable tool for the community.

This study is relevant for people working on the veterinary and medical aspects of schistosomiasis as well as on the biology of the schistosomes. It moves the field forward on showing that ongoing hybridization for livestock parasites is a reality. The methods used are appropriate. However, I have concerns regarding some affirmation in the introduction and the possible contamination of samples. More details are given below. I have also some other minor comments regarding needed information. Overall the manuscript is clearly written. When the comments will be addressed, the present study will be suitable for publication.

LEGEND

======

* p.: page

* §: paragraph

* l.: line

* Fig.: figure

* Tab.: table

This review is written in [markdown](https://en.wikipedia.org/wiki/Markdown).

**Part II – Major Issues: Key Experiments Required for Acceptance**

Reviewer #1: The f3 statistic is intended for populations, not individuals, so its use here is somewhat unconventional. It seems to me that it can indeed function as a test for hybridization when used in this way, but the authors should cite other examples of f3 being applied to individuals, or else explain that this is a novel use of the statistic and state any caveats. Allele frequency in a diploid individual can only be 0, 50%, or 100%, so essentially this is a test for an excess of heterozygous (50% frequency) SNPs among sites known to be fixed between S. curassoni and S. bovis.

Likewise, Patterson’s D is being used in a slightly unconventional way here. Patterson’s D is much better known as the ABBA-BABA test and it would aid reader comprehension to describe it as such. It is not a conventional test for hybridization. It is typically used to distinguish between incomplete lineage sorting and introgression, but that’s not what it’s doing here, so the authors should remove statements to that effect (e.g. Table 3, Figure 3, and “Genomic signatures of admixture” section). Instead, the authors are (correctly, in my view) rejecting the hypothesis of incomplete lineage sorting AND the hypothesis of introgression in favor if the hypothesis of hybridization. For the hybrids, D is near 0, which (absent any other information) would be consistent with incomplete lineage sorting, and certainly doesn’t rule out incomplete lineage sorting, though of course the ABBA BABA results are also consistent with the hybrid ancestries inferred from ADMIXUTRE and f3. I can’t see how Patterson’s D is formally testing any hypothesis here, but if it is the authors should explain the alternative hypothesis that is being ruled out.

Please provide more details about the F2 backcross inference and definition. Does “F2 backcross” mean a worm that has one purebred parent and one parent that is an F2, or a worm where both parents were derived from an F1 backcross? Specimen BK16_B8_11 is listed as an “F2 backcross” in Table 3, but there is no such designation in the NEWHYBRIDS results presented in Supp Data 8. What is the basis for inferring that it’s an F2 backcross and not an F2? In the text there are repeated mentions of “F2 backcrosses” (plural) but this seems to be the only example. Please clarify.

What does these results mean for the status of these taxa as species? How divergent are they? 309,040 fixed SNPs across a 397 Mb genome is fewer than would be expected between “good” species, but presumably many real SNPs were filtered out. Can the authors estimate the % nucleotide divergence between species based on the sites where reads did reliably align? On the other hand, it seems unlikely that the authors happened to stumble upon one of the first instances of gene flow between these taxa, so perhaps hybridization occurs often but there is selection against hybrids even if they are fertile.

Reviewer #2: None

Reviewer #3: MAJOR COMMENTS

My main concerns are below:

* In the introduction, the authors should probably be slightly more cautious when qualifying schistosomes of hybrids. Most "hybrids" have been identified with only single/few nuclear/mitochondrial markers. Recent studies showed that this can be misleading regarding when the hybridization took place and therefore the true status of hybrid. The authors discussed this point in their introduction and discussion. There are more specific comments below regarding some passages.

* While I recognize the effort of the authors to carefully assess possible contamination in their samples (p. 13 §2), I still have concerns:

- The authors focused on the same-species contamination but the inter-species contamination is the most problematic type because leading to identification of hybrids that were actually inter-species mating couples. Nothing is done to address this.

- Looking at the supplementary figure, some samples showed profile that could be problematic (for instance all the SHAE, SINT_01, SMAG_01, SMAT_01) but were not excluded. The criteria/algorithm for exclusion is unclear.

- There are methods relying on co-segregating SNPs on reads that are robust in estimating number of haplotypes in a sample. For instance [estMOI](https://academic.oup.com/bioinformatics/article/30/9/1292/235531), designed for malaria, could be applied on diploid organism, for which a number of 2 haplotypes would be expected. As this is reference agnostic, analyzing hybrids should not be a problem. Coupled with the current method, this should reveal intra and inter-species contamination. I would suggest to create some test cases with the current data and known pure samples for validating the approach.

Comments on the text

* **p. 9 l. 13-14**: "child in Niger was found to be infected with an hybrid schistosome derived from two livestock-infective schistosome species": this is based on single nuclear and mitochondrial markers. The authors should be more cautious and probably describe the results only based on the observation, especially knowing that no active hybridization has been identified in Niger at the genome level ([Platt et al. 2019](https://pubmed.ncbi.nlm.nih.gov/31251352/)). Here is a suggested rephrasing of the sentence: "child in Niger was found to be infected with a schistosome displaying markers derived from two livestock-infective schistosome species (...) futher suggesting ...".

* **p. 9 l. 22-23**: "hybridization between *S. bovis* and *S. haematobium* appears to have facilitated a 2013 outbreak of schistosomiasis in Corsica": a recent paper ([Rey et al.](https://journals.plos.org/plospathogens/article/file?id=10.1371/journal.ppat.1009313&type=printable)) reclassified the "hybrid" Corsican sample as a *S. haematobium*. The authors should therefore be cautious about considering this sample as hybrid. This also applies to **p. 10 l. 20-23**.

* **p. 10 l. 4**: "Hybrid schistosomes" should be "Schistosomes from possible hybridization event".

* **p. 10 l. 9**: "is likely" should be "could".

* **p. 10 l. 20**: please see previous comment. Again, caution is needed.

* **p. 11 l. 3-4**: "hybrid lineages are present" should be "hybrid lineages could be present".

* **p. 19 l. 9**: "readily": the authors should remove the adverb because the constraints leading to hybridization are unknown. While time since divergence can indeed explain how easy it is for these two species to hybridize, scarcity of partners within the host may in fact be the reason of the hybridization that may not occur otherwise.

* **p. 20 l. 5**: "hybrids combinations have been identified to infect humans": same remark as for p. 9 l. 13-14.

**Part III – Minor Issues: Editorial and Data Presentation Modifications**

Reviewer #1: Supp Figs 1 and 2 are essentially a single figure split in two, and likewise Supp Figs 3, 4, and 5 are essentially a single figure split into three. The captions are even repeated identically. I understand the need to spread the plots across multiple pages, but it’s confusing to have several figures that are described identically. If it has to be several figures, could they be organized by geography or host (e.g. all BK16_B8 samples in the same figure), so there is some justification for the split?

Supp Fig 7: MtDNA is inherited as a nonrecombining unit and thus its evolutionary history is fully represented by the phylogeny (part C). Since the variants are linked and not independent, it violates the assumptions of PCA and ADMIXTURE, so parts A, B, and D add nothing to the story and are arguably statistically invalid. A more robust analysis with be to add bootstraps to the phylogeny and/or to use a more powerful phylogeny inference method like maximum likelihood.

Reviewer #2: Please, include page and line numbers in any future submissions. It is highly annoying for a reviewer to comment on a paper without these numbers!

Figure 2C. Include bootstrap values

Some grammatical issues:

In Introduction: "Recently a child in Niger was found to be infected with an hybrid schistosome derived from two livestock-infective schistosome species.. "  'an hybrid' should probably be 'a hybrid'

In Introduction: there is something not quite right about the grammar in this sentence: "whereas more recent hybrid lineages could indicate demographic shifts or anthropogenic environmental changes influence parasite population structures."

In Discussion: 'In the same study, the authors did not find snails shedding S. bovis in Barkedji concluding that S. bovis was mostly likely imported from neighbouring regions.'  'mostly likely' should most likely be 'most likely'

In Discussion, conclusion: '..., supporting that indicated by recent population genetic approaches.' Check grammar.

Reviewer #3: MINOR COMMENTS

Comments on the text

General comment: I strongly encouraged the authors to add numbered lines for next submissions (can be done automatically with any word processor). This will greatly facilitate the reviewing process.

* **p. 6**: "spp" should be "spp." (period missing).

* **p. 10 l. 14**: "from infected children with Niger": this formulation sounds strange. Should this be "from infected children in Niger" ("with" replaced with "in")?

* **p. 12 l. 8**: "most recent": the version v7 is no longer the most recent as v9 is now available. The authors should replace this with "recent" or "version 7" (removing the "(v7)" in this case).

* **p. 25 and 26**: Genotyping and sequencing sections: the authors should provide the quantity of DNA used for genotyping and preparing the sequencing libraries. What was the kit used for library preparation?

* **p. 27 l. 20**: "S. curassoni" should be "*S. curassoni*" (italic missing).

* **p. 27 last l.**: "S. bovis" should be "*S. bovis*" (italic missing).

* **p. 43**: "published datasets." should be "published datasets)." (closing parenthesis missing).

Comments on the figures

* **Fig. 1 & 2**: letters associated to each panel should be either lower case or upper case for consistency.

* **Supp. Fig. 3 to 5**: only 3 out of 4 samples are tagged as excluded. I suspect that this is due to the removed sample showing low genotyping rate. This should be clarified in the text.

PLOS authors have the option to publish the peer review history of their article (what does this mean?). If published, this will include your full peer review and any attached files.

Reviewer #1: No

Reviewer #2: No

Reviewer #3: No

Figure Files:

Data Requirements:

Reproducibility:

References:

---

## [Editor Report · Decision Letter 1]

27 Jun 2022

Dear Mr Berger,

We are pleased to inform you that your manuscript 'Genomic evidence of contemporary hybridization between Schistosoma species' has been provisionally accepted for publication in PLOS Pathogens.

Best regards,

James J Collins III

Section Editor

PLOS Pathogens

James Collins III

Section Editor

PLOS Pathogens

Kasturi Haldar

Editor-in-Chief

PLOS Pathogens

orcid.org/0000-0001-5065-158X

Michael Malim

Editor-in-Chief

PLOS Pathogens

orcid.org/0000-0002-7699-2064
---

## [Editor Report · Acceptance letter]

1 Aug 2022

Dear Mr Berger,

We are delighted to inform you that your manuscript, "Genomic evidence of contemporary hybridization between *Schistosoma* species," has been formally accepted for publication in PLOS Pathogens.

Best regards,

Kasturi Haldar

Editor-in-Chief

PLOS Pathogens

orcid.org/0000-0001-5065-158X

Michael Malim

Editor-in-Chief

PLOS Pathogens

orcid.org/0000-0002-7699-2064